# Aetiology of vaginal discharge, urethral discharge, and genital ulcer in sub-Saharan Africa: A systematic review and meta-regression

Julia Michalow[1]*, Magdalene K. Walters[1]º, Olanrewaju Edun[1]º, Max Wybrant[1], Bethan Davies[2], Tendesayi Kufa[3], Thabitha Mathega[3], Sungai T. Chabata[4], Frances M. Cowan[4,5], Anne Cori[1], Marie-Claude Boily[1], Jeffrey W. Imai-Eaton[1,6]

1 MRC Centre for Global Infectious Disease Analysis, School of Public Health, Imperial College London, London, United Kingdom, 2 MRC Centre for Environment and Health, Department of Epidemiology and Biostatistics, School of Public Health, Imperial College London, London, United Kingdom, 3 Centre for HIV & STI, National Institute for Communicable Diseases, Johannesburg, South Africa, 4 Centre for Sexual Health and HIV AIDS Research (CeSHHAR), Harare, Zimbabwe, 5 Department of International Public Health, Liverpool School of Tropical Medicine, Liverpool, United Kingdom, 6 Center for Communicable Disease Dynamics, Department of Epidemiology, Harvard T.H. Chan School of Public Health, Boston, Massachusetts, United States of America

º These authors contributed equally to this work.

* j.michalow21@imperial.ac.uk

## Abstract

### Background

Syndromic management is widely used to treat symptomatic sexually transmitted infections in settings without aetiologic diagnostics. However, underlying aetiologies and consequent treatment suitability are uncertain without regular assessment. This systematic review estimated the distribution, trends, and determinants of aetiologies for vaginal discharge, urethral discharge, and genital ulcer in sub-Saharan Africa (SSA).

### Methods and findings

We searched Embase, MEDLINE, Global Health, Web of Science, and grey literature from inception until December 20, 2023, for observational studies reporting aetiologic diagnoses among symptomatic populations in SSA. We adjusted observations for diagnostic test performance, used generalised linear mixed-effects meta-regressions to generate estimates, and critically appraised studies using an adapted Joanna Briggs Institute checklist. Of 4,418 identified records, 206 reports were included from 190 studies in 32 countries conducted between 1969 and 2022. In 2015, estimated primary aetiologies for vaginal discharge were candidiasis (69.4% [95% confidence interval (CI): 44.3% to 86.6%], n = 50), bacterial vaginosis (50.0% [95% CI: 32.3% to 67.8%], n = 39), chlamydia (16.2% [95% CI: 8.6% to 28.5%], n = 50), and trichomoniasis (12.9% [95% CI: 7.7% to 20.7%], n = 80); for urethral discharge were gonorrhoea (77.1% [95% CI: 68.1% to 84.1%], n = 68) and chlamydia (21.9% [95% CI: 15.4% to 30.3%], n = 48); and for genital ulcer were herpes simplex virus

**Data Availability Statement:** Data extracted from included studies and used for analysis are available as supplementary material (S2 Appendix). Code reproducing the analysis is available from https://doi.org/10.5281/zenodo.10849676.

**Funding:** JM acknowledges funding from the Imperial College President's PhD Fund. JWI-E acknowledges funding from the Bill & Melinda Gates Foundation (INV-006733, INV-002606). JM, MKW, OE, AC, M-CB, and JWI-E acknowledge funding from the MRC Centre for Global Infectious Disease Analysis (reference MR/R015600/1), jointly funded by the UK Medical Research Council (MRC) and the UK Foreign, Commonwealth & Development Office (FCDO), under the MRC/FCDO Concordat agreement and is also part of the EDCTP2 programme supported by the European Union. Under the grant conditions of UKRI and the Bill & Melinda Gates Foundation, a Creative Commons Attribution 4.0 Generic License (CC BY) has already been assigned to any Author Accepted Manuscript version arising from this submission. The funders had no role in study design, data collection and analysis, decision to publish, or preparation of the manuscript.

**Competing interests:** The authors have declared that no competing interests exist.

**Abbreviations:** aOR, adjusted odds ratio; BV, bacterial vaginosis; CA, *Candida albicans*; CI, confidence interval; CS, *Candida* species; CT, *Chlamydia trachomatis*; GU, genital ulcer; HD, *Haemophilus ducreyi*; HSV-1, Herpes simplex virus type 1; HSV-2, herpes simplex virus type 2; LGV, lymphogranuloma venereum; MG, *Mycoplasma genitalium*; NAAT, nucleic acid amplification testing; NG, *Neisseria gonorrhoeae*; RTI, reproductive tract infection; SSA, sub-Saharan Africa; STI, sexually transmitted infection; TP, *Treponema pallidum*; TV, *Trichomonas vaginalis*; UD, urethral discharge; VD, vaginal discharge; WHO, World Health Organization.

type 2 (HSV-2) (48.3% [95% CI: 32.9% to 64.1%], $n = 47$) and syphilis (9.3% [95% CI: 6.4% to 13.4%], $n = 117$). Temporal variation was substantial, particularly for genital ulcer where HSV-2 replaced chancroid as the primary cause. Aetiologic distributions for each symptom were largely the same across regions and population strata, despite HIV status and age being significantly associated with several infection diagnoses. Limitations of the review include the absence of studies in 16 of 48 SSA countries, substantial heterogeneity in study observations, and impeded assessment of this variability due to incomplete or inconsistent reporting across studies.

## Conclusions

In our study, syndrome aetiologies in SSA aligned with World Health Organization guidelines without strong evidence of geographic or demographic variation, supporting broad guideline applicability. Temporal changes underscore the importance of regular aetiologic re-assessment for effective syndromic management.

## PROSPERO number

CRD42022348045.

---

## Author summary

### Why was this study done?

- Symptom-based case management is common for treating sexually transmitted infections (STIs) in sub-Saharan Africa (SSA).

- Characterising the infectious aetiologies (causes) of each syndrome is crucial to ensure adequate choice of treatment.

- Recent comprehensive assessments on the aetiologies for vaginal discharge, urethral discharge, and genital ulcer are lacking in SSA.

### What did the researchers do and find?

- We conducted a systematic review that included 190 studies in 32 SSA countries spanning 1969 and 2022.

- We accounted for the sensitivity and specificity of different diagnostic tests used across studies and used meta-regression models to estimate the distribution of infections causing each symptom.

- We determined that the main aetiologies for vaginal discharge were candidiasis (69% of cases in 2015), bacterial vaginosis (50%), chlamydia (16%), and trichomoniasis (13%); for urethral discharge were gonorrhoea (77%) and chlamydia (22%); and for genital ulcer were herpes simplex virus type 2 (HSV-2) (48%) and syphilis (9%).

- Distributions of infectious aetiologies were similar across regions and population sub-groups but changed over time.

### What do these findings mean?

- The findings support the applicability and continued use of World Health Organization guidelines for symptomatic STI management across SSA settings.

- National STI programmes should re-assess aetiologies regularly due to changes over time.

- Limitations of the review include that no data were available in 16 of 48 SSA countries, there was large variability in results across studies identified in the review, and certain information of interest was missing or inconsistently reported across studies.

## Introduction

In 2020, the World Health Organization (WHO) estimated 374 million new infections worldwide of the 4 most common curable sexually transmitted infections (STIs): chlamydia, gonorrhoea, syphilis, and trichomoniasis [1]. In sub-Saharan Africa (SSA), which has 40% of the global STI burden [2], access to laboratory or point-of-care aetiologic diagnostics is limited. Syndromic case management, in which probable causative infection(s) are treated based on presenting symptoms, was introduced by WHO in 1984 and remains the standard of care [3]. The approach enables rapid treatment but has several limitations. Many STIs remain untreated partly due to high rates of asymptomatic infection, particularly in women [4]. The diagnostic accuracy of syndromic management algorithms is suboptimal, despite efforts to incorporate evolving STI epidemiology [5]. WHO guidance recommends national re-assessment of syndrome aetiologies every 2 years to ensure the relevance of syndromic algorithms [6,7], but only 40% of African countries (11/26 in WHO survey) reported including these assessments in STI surveillance [8]. These factors collectively may lead to unnecessary, incorrect, and missed treatment for STIs, which is particularly concerning amid rising antimicrobial resistance [5].

In 2021, the WHO released new guidelines for symptomatic STI management, the first update since 2003 [3]. These updates were informed by global systematic reviews of studies assessing the diagnostic performance of different syndromic management algorithms. Although the guidelines accounted for changes over time in the underlying causes of each syndrome, they lacked a comprehensive review and quantification of the STI distribution among symptomatic populations in different geographic areas. This information would support future modification of syndromic management protocols to align with local epidemiology.

The objective of our study was to characterise the aetiologies for 3 prevalent STI symptoms in SSA: vaginal discharge, urethral discharge, and genital ulcer. We performed a systematic review and meta-analysis to estimate the distribution of aetiologies for each symptom, investigate their spatiotemporal changes, and evaluate variation according to population-specific determinants, namely sex, HIV status, and age.

## Methods

### Data sources and search strategy

We systematically searched for studies assessing the aetiology of vaginal discharge, urethral discharge, and genital ulcer in SSA. Embase (Ovid), MEDLINE (Ovid), Global Health (Ovid), and Web of Science were searched from database inception to 20 December 2023. Search term domains included relevant terms and synonyms for "symptoms," "infections," and "sub-Saharan Africa" (Table A in S1 Appendix). We performed a comprehensive grey literature search up to 20 December 2023 using the same search term domains. Sources included websites and reports by the WHO, UNAIDS, and Ministries of Health, and conference abstracts published between 2000 and 2023 from the STI & HIV World Congress, International AIDS Society, and International Conference on AIDS and STIs in Africa. We contacted authors of relevant conference abstracts to enquire about potential unpublished data. Sub-Saharan Africa and its subregions were defined according to the UN M49 standard (Table B in S1 Appendix) [9].

### Study selection and eligibility criteria

Search results were uploaded and de-duplicated in Covidence systematic review software (Veritas Health Innovation, Melbourne, Australia). Two reviewers independently screened titles and abstract records for eligibility, and then assessed full-text reports for inclusion. Any discrepancies were resolved through consensus or by a third reviewer.

We included reports that contained empirical data on the proportion of women with vaginal discharge (VD) diagnosed with bacterial vaginosis (BV), any *Candida* species (CS), *Candida albicans* (CA), *Chlamydia trachomatis* (CT), *Mycoplasma genitalium* (MG), *Neisseria gonorrhoeae* (NG), *Trichomonas vaginalis* (TV), or unknown aetiology (negative aetiologic test results); proportion of men with urethral discharge (UD) diagnosed with CT, MG, NG, TV, or unknown aetiology; and proportion of men and women with genital ulcer (GU) diagnosed with *Haemophilus ducreyi* (HD), herpes simplex virus of unspecified type (HSV) or types 1 or 2 (HSV-1 or HSV-2), lymphogranuloma venereum (LGV) caused by CT serovars L1–L3, *Treponema pallidum* (TP), or unknown aetiology. Since BV and candidiasis are not considered STIs, but are common causes of vaginal discharge, we refer to the broader term reproductive tract infections (RTIs) in this study.

Studies were included if: (1) there were participants symptomatic at the time of testing, defined by the presence of either self-reported or clinician-evaluated abnormal vaginal discharge, urethral discharge, or genital ulcer; (2) participants were aged 10 years and older; (3) the sample size was at least 10; and (4) the diagnostic methodology for each infection was described and assessed as valid. While our primary interest was studies among sexually active adult populations, we used a minimum age of 10 years to avoid excluding studies which enrolled both eligible sexually active adolescent and adult participants; in such studies ($N = 18$), the majority of participants were adults. Diagnostic method validity was assessed by cross-referencing its inclusion in either current or previous published recommendations [3,10–15]. Exclusion criteria were: (1) qualitative studies, case reports, commentaries, reviews, mathematical modelling studies, and longitudinal and randomised controlled studies reporting outcomes post-baseline only; and (2) studies published in languages other than English, French, or Portuguese.

### Data extraction

Data were independently double extracted from each study (number of studies denoted as $N$) with discrepancies resolved through consensus or by a third reviewer. A study observation

(number of observations across all studies denoted as *n*) was the proportion of symptomatic individuals diagnosed with a given RTI or, if not directly reported, the numerator and denominator to calculate the proportion. In cases of RTI coinfection, we extracted each RTI separately, potentially causing the total infected proportion to exceed 1 when aggregating all observations for a given population. We extracted observations stratified by population type (symptomatic clinic attendees; general populations; higher-risk general populations, such as truck drivers, mineworkers, soldiers, and bar workers; or key populations, such as sex workers, and men who have sex with men), country, year or time-period if not stratified by year, sex, HIV status, and age group as available. We excluded strata subsamples of fewer than 10 participants. Information on study characteristics, participant characteristics, and diagnostic methods was also extracted (Table C in S1 Appendix). We prepared and extracted unpublished data from 2 databases identified during the search (Text A and Table D in S1 Appendix).

If multiple reports included the same outcome(s) for a study, we preferentially retained observations from the largest sample or, if samples were the same size, observations from the report with the largest number of RTIs tested. If the sample sizes and number of RTIs were equal, the most recent report was retained. When available, we used observations tabulated directly from databases and excluded corresponding published articles to avoid duplication.

If a study reported outcomes for self-reported and clinician-evaluated symptoms in the same population, we preferentially extracted observations for the latter. If multiple diagnostic tests were conducted, outcomes based on the most accurate test method for the pathogen and symptom (Tables E, F, and G in S1 Appendix) were preferentially extracted. When multiple diagnostic tests were used for syphilis among those with genital ulcer, we preferentially extracted observations for tests using ulcer swab specimens over serology [3]. When syphilis serology was used, we prioritised observations from the combination of non-treponemal and treponemal tests when available. The proportion with "unknown aetiology" was only extracted from studies with observations for 3 or more RTI pathogens, regardless of the specific pathogens tested.

## Data analysis

To adjust reported proportions for diagnostic test performance, we classified diagnostic tests into broad categories and compiled sensitivity and specificity estimates for each test category from literature, with priority given to characteristics published by the WHO (Text B and Tables E, F, and G in S1 Appendix) [10,16]. We used a Bayesian approach to estimate the true proportion of symptomatic individuals with a given RTI [17,18]. Adjusted numerators and denominators were calculated based on the mean and standard error of the true proportion and used to pool observations (Text B in S1 Appendix).

The diagnosed proportion for each infection was estimated using observations among adults of mixed or unmeasured HIV status. We estimated time trends in the diagnosed proportion by region via generalised linear mixed-effects meta-regressions for each symptom [19]. Models were specified a priori to include fixed effects for RTI, the interaction of RTI and year (midpoint date of data collection measured as continuous calendar year), and the interaction of RTI and sex (genital ulcer only), random intercepts and slopes per year for the interaction of RTI and region (central and western, eastern, or southern Africa), and observation-level random intercepts to account for between-study heterogeneity. Pooled regional means were weighted by sex-matched regional population estimates for adults 15 years and older in 2015 from the UN World Population Prospects 2022 [20].

To assess the effects of HIV status and age, we used observations reported by these stratifications. We extended the meta-regressions described above to include fixed effects for either

HIV status (HIV–positive, HIV–negative) or age group (<25 years, ≥25 years), but did not include random slopes due to their low standard deviation in the main analysis.

We adapted the Joanna Briggs Institute critical appraisal tool for prevalence studies to assess study design (objective of the study), selection bias (clarity of inclusion criteria, appropriateness of recruitment method, adequacy of participation, and detail of participant characterisation), measurement bias (objectivity in symptom definition, consistency of diagnostic methodology, and avoidance of misclassified results), precision (sufficiency of sample size), and reporting quality (ambiguity of results) (Table H in S1 Appendix) [21]. Each of the 10 criteria was independently double assessed for each report, with discrepancies resolved through consensus or by a third reviewer. We assessed the association of these criteria with the RTI proportions by extending the meta-regressions to include fixed effects for each criterion.

To assess sensitivity to the diagnostic test adjustments, we compared trend estimates between models using observations as reported and models using observations adjusted for diagnostic test performance. This comparison was conducted for all diagnostic test types (all RTIs) and separately for nucleic acid amplification testing (NAAT) (all RTIs, excluding BV, CA, and CS).

Unless stated otherwise, all analyses use observations adjusted for diagnostic test performance. Studies that collected data from countries in multiple regions, but did not report data stratified by country or region, were classified using the region from which most participants were recruited. Primary meta-regression pooled results are reported as model predictions for the year 2015, representing the most recent quinquennium within the timespan of substantial available data. Results are presented as means with 95% confidence intervals (95% CIs). Model coefficients are presented as adjusted odds ratios (aORs), with CIs calculated on the log odds scale before exponentiation. Study observation heterogeneity is assessed per model as the percentage of total variance attributed to observation-level random effects [22]. Analyses were conducted in R version 4.2.3, Bayesian adjustments for diagnostic test performance were implemented in rstan version 2.26.21 [23], and mixed-effects logistic meta-regression models were implemented in glmmTMB version 1.1.8 [24].

This systematic review and meta-analysis was pre-registered on PROSPERO (CRD: 42022348045) [25] and reported as per the Preferred Reporting Items for Systematic Reviews and Meta-Analyses (PRISMA) guideline (S1 Checklist) [26]. The study protocol was reviewed and approved by the Imperial College Research Ethics Committee (ICREC #6389606).

## Results

### Search results and scope

We identified 7,780 records through the database search, of which 3,362 were duplicates and 4,418 were screened (Fig 1). Of these, 745 full-text publications were assessed for eligibility and 199 were included. We further identified 40 records through grey literature sources and citation searching, of which 5 full-text publications and 2 databases were assessed for eligibility and included. The 2 included databases were from the National Institute for Communicable Diseases periodic aetiologic prevalence surveys among symptomatic primary healthcare clinic attendees in South Africa between 2006 and 2022 [27–29], and the Centre for Sexual Health and HIV/AIDS Research female sex worker population size estimation study in Zimbabwe in 2017 [30].

Overall, 206 reports were included from 190 independent studies (number of studies per symptom (N): $N_{VD} = 87$, $N_{UD} = 55$, $N_{GU} = 80$) spanning 1969 to 2022 (Table 1 and S2 Appendix). Of these, 166 studies focused on a single symptom, 16 studies on 2 symptoms, and 8 studies on all 3 symptoms. Studies were conducted in 32 of 48 SSA countries included in our

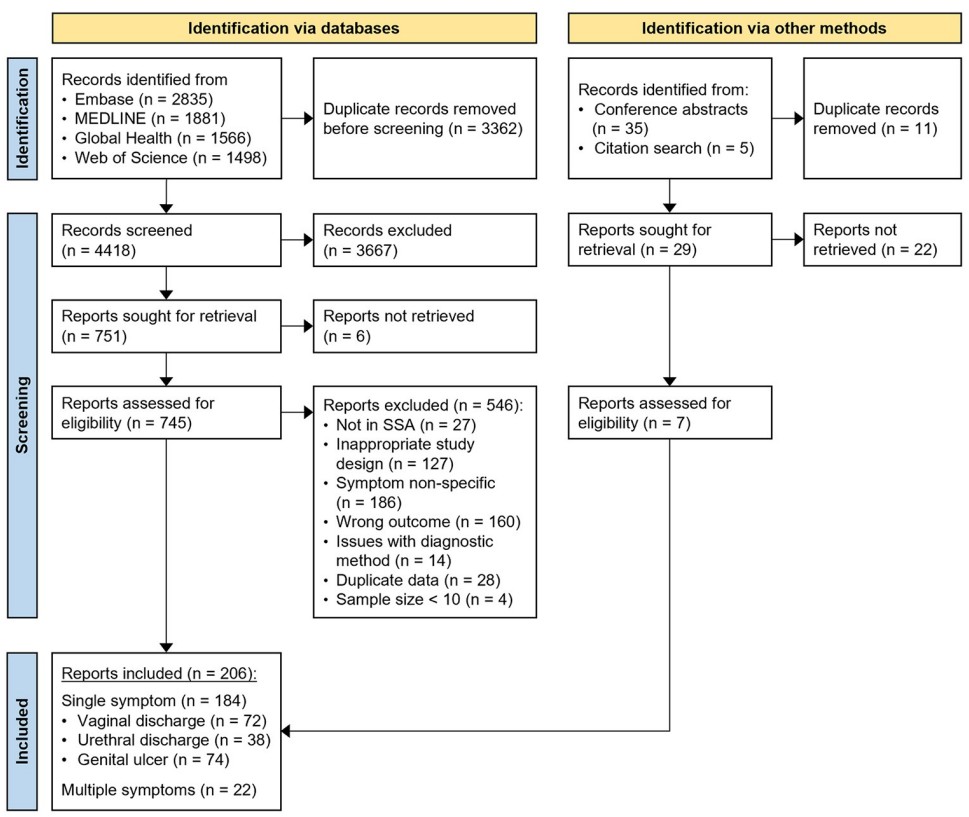

**Fig 1. Study selection flowchart.** Records include titles and abstracts identified for initial screening. Reports include full-text published articles or databases assessed for inclusion.

search, with a median of 5 studies per country. Most studies were in eastern Africa ($N_{VD}$ = 35/87, $N_{UD}$ = 32/55, $N_{GU}$ = 50/80) and few were in central Africa ($N_{VD}$ = 4/87, $N_{UD}$ = 2/55, $N_{GU}$ = 2/80). Studies were predominantly in South Africa ($N_{VD}$ = 13/87, $N_{UD}$ = 10/55, $N_{GU}$ = 18/80), along with Nigeria ($N_{VD}$ = 15/87) for vaginal discharge and Kenya ($N_{GU}$ = 18/80) for genital ulcer (Fig A in S1 Appendix). Most studies occurred after 2000 ($N_{VD}$ = 61/87, $N_{UD}$ = 28/55) for vaginal discharge and urethral discharge and between 1990 and 1999 ($N_{GU}$ = 36/80) for genital ulcer. Studies were predominantly among symptomatic clinic attendees ($N_{VD}$ = 60/87, $N_{UD}$ = 49/55, $N_{GU}$ = 63/80) and among individuals of mixed HIV status ($N_{VD}$ = 32/87, $N_{UD}$ = 19/55, $N_{GU}$ = 51/80). Few studies reported the prevalence of HIV ($N_{VD}$ = 22/87, $N_{UD}$ = 11/55, $N_{GU}$ = 44/80) or the mean or median age ($N_{VD}$ = 26/87, $N_{UD}$ = 17/55, $N_{GU}$ = 22/80) of participants. Studies were mostly cross-sectional ($N_{VD}$ = 73/87, $N_{UD}$ = 46/55, $N_{GU}$ = 63/80). Most had sample sizes of 100 of more ($N_{VD}$ = 61/87, $N_{UD}$ = 39/55, $N_{GU}$ = 46/80) and tested for more than one pathogen ($N_{VD}$ = 57/87, $N_{UD}$ = 35/55, $N_{GU}$ = 57/80). The most frequently assessed aetiologies were TV ($N_{VD}$ = 60/87) and NG ($N_{VD}$ = 42/87) for vaginal discharge; NG ($N_{UD}$ = 50/55), CT ($N_{UD}$ = 31/55), and TV ($N_{UD}$ = 28/55) for urethral discharge; and TP ($N_{GU}$ = 68/80) and HD ($N_{GU}$ = 58/80) for genital ulcer. Study characteristics varied by sub-analysis (Tables I, J, and K in S1 Appendix).

## Regional trends in the aetiology of RTI symptoms

In 2015, CS and BV, rather than STIs, were the primary aetiologies for vaginal discharge (Fig 2A and Table 2A). The proportion of VD cases with CS was 69.4% (95% CI: 44.3% to 86.6%;

**Table 1. Summary of participant and study characteristics for included studies.**

| | | Vaginal discharge (N_VD = 87) | Urethral discharge (N_UD = 55) | Genital ulcer (N_GU = 80) | All symptoms (N = 190) |
|---|---|---|---|---|---|
| **Population characteristics** | | | | | |
| **Population group**[1] | Symptomatic clinic attendee | 60 (69%) | 49 (89.1%) | 63 (78.8%) | 145 (76.3%) |
| | General | 25 (28.7%) | 2 (3.6%) | 6 (7.5%) | 32 (16.8%) |
| | Higher-risk general | 2 (2.3%) | 3 (5.5%) | 7 (8.8%) | 10 (5.3%) |
| | Key populations | 6 (6.9%) | 1 (1.8%) | 8 (10%) | 14 (7.4%) |
| | Youth | 16 (18.4%) | 6 (10.9%) | 6 (7.5%) | 21 (11.1%) |
| **Sex** | Male | - | 55 (100.0%) | 20 (25.0%) | 56 (29.5%) |
| | Female | 87 (100.0%) | - | 21 (26.3%) | 86 (45.3%) |
| | Mixed | - | - | 39 (48.8%) | 48 (25.3%) |
| **Mean or median age** | <25 years | 8 (9.2%) | 2 (3.6%) | 1 (1.3%) | 8 (4.2%) |
| | ≥25 years | 18 (20.7%) | 15 (27.3%) | 21 (26.3%) | 49 (25.8%) |
| | NR | 61 (70.1%) | 38 (69.1%) | 58 (72.5%) | 133 (70%) |
| **HIV status** | Positive | 1 (1.1%) | 1 (1.8%) | 3 (3.8%) | 5 (2.6%) |
| | Negative | 4 (4.6%) | 1 (1.8%) | 1 (1.3%) | 5 (2.6%) |
| | Mixed | 32 (36.8%) | 19 (34.5%) | 51 (63.8%) | 81 (42.6%) |
| | NR | 50 (57.5%) | 34 (61.8%) | 25 (31.3%) | 99 (52.1%) |
| **HIV prevalence** | < 25% | 12 (13.8%) | 4 (7.3%) | 5 (6.3%) | 19 (10%) |
| | ≥ 25% | 10 (11.5%) | 7 (12.7%) | 39 (48.8%) | 46 (24.2%) |
| | NR | 65 (74.7%) | 44 (80.0%) | 36 (45.0%) | 125 (65.8%) |
| **Study characteristics** | | | | | |
| **Region**[1,2] | Central Africa | 4 (4.6%) | 2 (3.6%) | 2 (2.5%) | 7 (3.7%) |
| | Eastern Africa | 35 (40.2%) | 32 (58.2%) | 50 (62.5%) | 93 (48.9%) |
| | Southern Africa | 14 (16.1%) | 11 (20%) | 23 (28.8%) | 45 (23.7%) |
| | Western Africa | 33 (37.9%) | 9 (16.4%) | 8 (10%) | 47 (24.7%) |
| | Multiple (not stratified) | 1 (1.1%) | 1 (1.8%) | 1 (1.3%) | 2 (1.1%) |
| **Study midpoint year**[1,3] | <1990 | 8 (9.2%) | 10 (18.2%) | 18 (22.5%) | 33 (17.4%) |
| | 1990–1999 | 21 (24.1%) | 18 (32.7%) | 36 (45.0%) | 63 (33.2%) |
| | 2000–2009 | 28 (32.2%) | 15 (27.3%) | 19 (23.8%) | 52 (27.4%) |
| | 2010–2023 | 33 (37.9%) | 13 (23.6%) | 11 (13.8%) | 48 (25.3%) |
| **Study design** | Cross sectional | 73 (83.9%) | 46 (83.6%) | 63 (78.8%) | 157 (82.6%) |
| | Cohort baseline | 11 (12.6%) | 8 (14.5%) | 12 (15%) | 25 (13.2%) |
| | RCT baseline | 3 (3.4%) | 1 (1.8%) | 5 (6.3%) | 8 (4.2%) |
| **Sampling method** | Non-probability based | 72 (82.8%) | 51 (92.7%) | 73 (91.3%) | 169 (88.9%) |
| | Probability based | 9 (10.3%) | 3 (5.5%) | 6 (7.5%) | 13 (6.8%) |
| | NR | 6 (6.9%) | 1 (1.8%) | 1 (1.3%) | 8 (4.2%) |
| **Sample size**[4] | <100 | 26 (29.9%) | 16 (29.1%) | 34 (42.5%) | 68 (35.8%) |
| | ≥100 | 61 (70.1%) | 39 (70.9%) | 46 (57.5%) | 122 (64.2%) |

(*Continued*)

**Table 1.** (Continued)

| | | Vaginal discharge ($N_{VD}$ = 87) | Urethral discharge ($N_{UD}$ = 55) | Genital ulcer ($N_{GU}$ = 80) | All symptoms ($N$ = 190) |
|---|---|---|---|---|---|
| **Pathogen detected[1]** | BV | 28 (32.2%) | - | - | 28 (14.7%) |
| | CA | 9 (10.3%) | - | - | 9 (4.7%) |
| | CS | 34 (39.1%) | - | - | 34 (17.9%) |
| | CT | 35 (40.2%) | 31 (56.4%) | - | 56 (29.5%) |
| | HD | - | - | 58 (72.5%) | 58 (30.5%) |
| | HSV | - | - | 31 (38.8%) | 31 (16.3%) |
| | HSV-1 | - | - | 15 (18.8%) | 15 (7.9%) |
| | HSV-2 | - | - | 23 (28.8%) | 23 (12.1%) |
| | LGV | - | - | 11 (13.8%) | 11 (5.8%) |
| | MG | 7 (8.0%) | 8 (14.5%) | - | 13 (6.8%) |
| | NG | 42 (48.3%) | 50 (90.9%) | - | 78 (41.1%) |
| | TP | - | - | 68 (85.0%) | 68 (35.8%) |
| | TV | 60 (69.0%) | 28 (50.9%) | - | 77 (40.5%) |
| | None[5] | 15 (17.2%) | 12 (21.8%) | 39 (48.8%) | 59 (31.1%) |
| **Number pathogens extracted per study** | 1 | 30 (34.5%) | 20 (36.4%) | 23 (28.8%) | 57 (30.0%) |
| | 2–3 | 29 (33.3%) | 27 (49.1%) | 40 (50.0%) | 85 (44.7%) |
| | 4–6 | 28 (32.2%) | 8 (14.5%) | 17 (21.3%) | 48 (25.3%) |

$N$ = number of studies.

[1]The same study is included in more than 1 subcategory when it reports across different variable levels or multiple variable levels are relevant.

[2]Regions classified according to UN M49 Standard. Central Africa: Angola, Cameroon, Central African Republic, Chad, Congo, Democratic Republic of Congo, Equatorial Guinea, Gabon, São Tomé and Príncipe. Eastern Africa: Burundi, Comoros, Djibouti, Eritrea, Ethiopia, Kenya, Madagascar, Malawi, Mauritius, Mozambique, Rwanda, Seychelles, Somalia, South Sudan, Uganda, United Republic of Tanzania, Zambia, Zimbabwe. Southern Africa: Botswana, Eswatini, Lesotho, Namibia, South Africa. Western Africa: Benin, Burkina Faso, Cabo Verde, Cote d'Ivoire, Gambia, Ghana, Guinea-Bissau, Guinea, Liberia, Mali, Mauritania, Niger, Nigeria, Senegal, Sierra Leone, Togo.

[3]Mid-point year between start and end dates of data collection. If dates were not reported, year of publication was used as proxy.

[4]Sample size defined as the number of symptomatic individuals tested in the study. If a study used different samples across tests, the largest sample size is reported.

[5]Unknown aetiology reported for studies with 3 or more pathogens extracted.

BV: Bacterial vaginosis, CA: *Candida albicans*, CS: *Candida* species (any), CT: *Chlamydia trachomatis*, HD: *Haemophilus ducreyi*, HSV: Herpes simplex virus (unspecified), HSV-1: Herpes simplex virus type 1, HSV-2: Herpes simplex virus type 2, LGV: lymphogranuloma venereum, MG: *Mycoplasma genitalium*, NG: *Neisseria gonorrhoeae*, TP: *Treponema pallidum*, TV: *Trichomonas vaginalis*. None: unknown aetiology. NR: Not reported.

number of observations per symptom ($n$): $n_{VD}$ = 50), with BV was 50.0% (95% CI: 32.3% to 67.8%, $n_{VD}$ = 39), and with CA was 31.5% (95% CI: 12.4% to 59.9%, $n_{VD}$ = 9). CT (16.2% [95% CI: 8.6% to 28.5%], $n_{VD}$ = 50) and TV (12.9% [95% CI: 7.7% to 20.7%], $n_{VD}$ = 80) were the most prominent STIs, while a relatively low proportion of cases were due to NG (6.6% [95% CI: 3.3% to 12.7%], $n_{VD}$ = 63) and MG (5.4% [95% CI: 1.8% to 14.8%], $n_{VD}$ = 20). A high proportion (25.1% [95% CI: 10.7% to 48.4%], $n_{VD}$ = 28) of cases did not have an identified aetiology, despite apparent high levels of coinfection. Diagnosed proportions increased over time for CS (aOR per year: 1.10 [95% CI: 1.05 to 1.15]), decreased for TV (aOR: 0.94 [95% CI: 0.91 to 0.97]), and were relatively constant for other RTIs (Table L in S1 Appendix). There were limited data to assess temporal trends for MG, for which most studies were conducted after 2005.

For urethral discharge, NG has remained the primary aetiology from the 1970s to present (Fig 2B and Table 2B). In 2015, NG was diagnosed in 77.1% (95% CI: 68.1% to 84.1%, $n_{UD}$ =

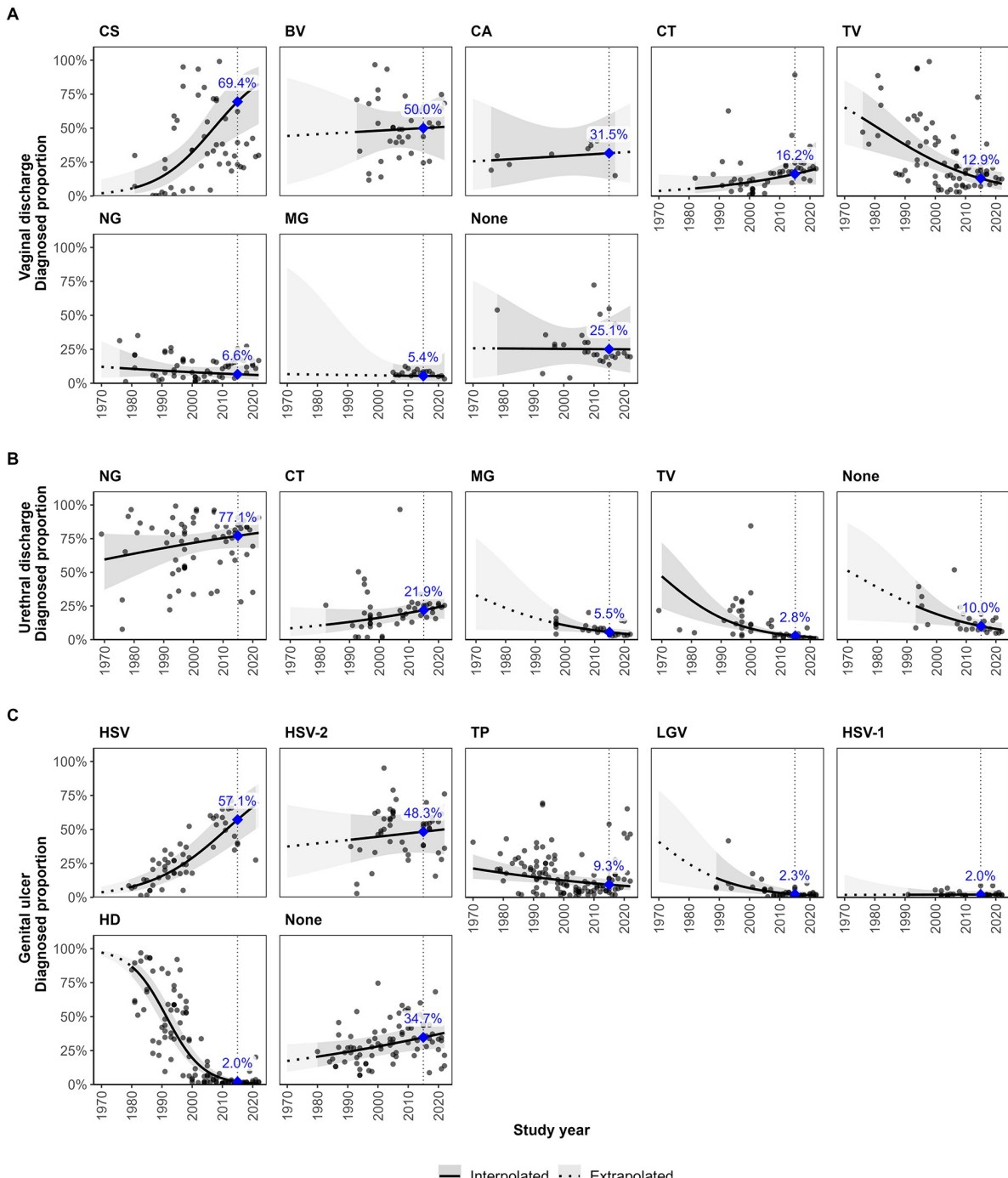

**Fig 2. Estimated diagnosed proportion per pathogen over time among symptomatic adults of mixed or unmeasured HIV status in sub-Saharan Africa.** Diagnosed proportion per pathogen among adults symptomatic with (A) vaginal discharge, (B) urethral discharge, and (C) genital ulcer. Proportions estimated using generalised linear mixed-effects model for each symptom with fixed effects for RTI and the interaction of RTI with year and sex (genital ulcer only), random intercepts and slopes per year for the interaction of RTI and region, and observation-level random intercepts. Lines and shaded areas represent sex-matched population-weighted mean proportions and 95% CIs. Solid lines and darker shading denote estimates and confidence intervals within the observed data range (interpolated), while dotted lines and lighter shading indicate estimates and confidence intervals beyond that time frame (extrapolated). Blue points and labels represent population-weighted mean proportions in 2015. Vertical dotted lines are through the year 2015. Grey points represent study observations adjusted for diagnostic test performance. BV: Bacterial vaginosis, CA: *Candida albicans*, CS: *Candida* species (any), CT: *Chlamydia trachomatis*, HD: *Haemophilus ducreyi*, HSV: Herpes simplex virus (unspecified), HSV-1: Herpes simplex virus type 1, HSV-2: Herpes simplex virus type 2, LGV: lymphogranuloma venereum, MG: *Mycoplasma genitalium*, NG: *Neisseria gonorrhoeae*, TP: *Treponema pallidum*, TV: *Trichomonas vaginalis*, None: unknown aetiology.

**Table 2. Estimated diagnosed proportion per pathogen in sub-Saharan Africa in 2015.**

| Pathogen | Overall | | | HIV status | | | | | | Age group | | | | | |
| --- | --- | --- | --- | --- | --- | --- | --- | --- | --- | --- | --- | --- | --- | --- | --- |
| | | | | HIV positive | | | HIV negative | | | < 25years | | | ≥25 years | | |
| | $N_s$ | $N_o$ | % (95% CI) | $N_s$ | $N_o$ | % (95% CI) | $N_s$ | $N_o$ | % (95% CI) | $N_s$ | $N_o$ | % (95% CI) | $N_s$ | $N_o$ | % (95% CI) |
| **A. Vaginal discharge** | | | | | | | | | | | | | | | |
| CS | 31 | 50 | 69.4 (44.3–86.6) | 4 | 17 | 31.4 (19.9–45.8) | 4 | 17 | 47.9 (34.2–61.9) | 5 | 18 | 51.1 (32.1–69.7) | 6 | 19 | 36.4 (20.9–55.4) |
| BV | 23 | 39 | 50.0 (32.3–67.8) | 6 | 20 | 45.5 (34.4–56.9) | 8 | 22 | 31.1 (22.3–41.4) | 6 | 21 | 39.5 (27.7–52.7) | 7 | 21 | 36.8 (25.2–50.0) |
| CA | 9 | 9 | 31.5 (12.4–59.9) | 0 | 0 | - | 0 | 0 | - | 2 | 2 | 51.8 (14.6–87.1) | 2 | 2 | 33.3 (7.7–74.8) |
| CT | 30 | 50 | 16.2 (8.6–28.5) | 5 | 19 | 10.2 (6.1–16.5) | 8 | 22 | 10.7 (6.7–16.8) | 6 | 20 | 11.6 (5.5–22.9) | 4 | 18 | 5.2 (2.4–11.1) |
| TV | 55 | 80 | 12.9 (7.8–20.7) | 8 | 22 | 17.1 (11.7–24.4) | 9 | 23 | 8.8 (5.9–12.9) | 11 | 25 | 6.3 (3.7–10.4) | 11 | 25 | 7.3 (4.4–12.1) |
| NG | 38 | 63 | 6.6 (3.3–12.7) | 6 | 20 | 13.5 (8.4–20.8) | 7 | 21 | 8.8 (5.5–13.9) | 7 | 21 | 5.1 (2.5–10.3) | 8 | 22 | 3.2 (1.5–6.5) |
| MG | 5 | 20 | 5.4 (1.8–14.8) | 3 | 17 | 8.2 (4.6–14.3) | 5 | 19 | 3.7 (2.1–6.6) | 1 | 15 | 6.7 (1.9–21.3) | 2 | 16 | 3.5 (1.0–11.7) |
| None | 13 | 28 | 25.1 (10.7–48.4) | 3 | 17 | 13.8 (7.4–24.5) | 5 | 19 | 18.2 (10.2–30.2) | 4 | 18 | 20.6 (9.0–40.3) | 4 | 18 | 27.8 (13.0–49.9) |
| **B. Urethral discharge** | | | | | | | | | | | | | | | |
| NG | 47 | 68 | 77.1 (68.1–84.1) | 5 | 19 | 83.7 (79.3–87.3) | 4 | 19 | 82.9 (78.5–86.5) | 5 | 19 | 83.9 (79.7–87.4) | 3 | 17 | 79.0 (74.5–82.8) |
| CT | 27 | 48 | 21.9 (15.4–30.3) | 5 | 19 | 17.8 (14.0–22.4) | 4 | 19 | 28.5 (23.2–34.5) | 3 | 17 | 31.0 (25.4–37.1) | 1 | 15 | 22.0 (17.9–26.8) |
| MG | 8 | 29 | 5.5 (3.7–8.3) | 2 | 16 | 6.1 (4.4–8.3) | 2 | 17 | 5.8 (4.4–7.7) | 1 | 15 | 5.3 (3.8–7.3) | 1 | 15 | 5.2 (4.0–6.8) |
| TV | 25 | 46 | 2.8 (1.8–4.3) | 4 | 18 | 3.6 (2.4–5.4) | 2 | 17 | 2.0 (1.3–3.0) | 5 | 19 | 3.4 (2.3–5.1) | 2 | 16 | 2.6 (1.8–3.6) |
| None | 11 | 26 | 10.0 (6.8–14.5) | 3 | 17 | 11.1 (8.5–14.5) | 1 | 16 | 8.2 (6.2–10.8) | 1 | 15 | 6.8 (5.0–9.2) | 1 | 15 | 9.0 (7.1–11.5) |
| **C. Genital ulcer** | | | | | | | | | | | | | | | |
| HSV | 28 | 55 | 57.1 (40.9–71.9) | 11 | 29 | 56.8 (42.9–69.6) | 8 | 24 | 42.5 (29.4–56.7) | 1 | 12 | 55.1 (42.5–67.1) | 1 | 15 | 54.8 (43.3–65.9) |
| HSV-2 | 22 | 47 | 48.3 (32.9–64.1) | 14 | 29 | 52.4 (41.0–63.6) | 13 | 28 | 40.3 (29.8–51.9) | 3 | 11 | 43.2 (31.2–56.0) | 4 | 17 | 42.8 (32.4–53.9) |
| TP | 64 | 117 | 9.3 (6.4–13.4) | 22 | 57 | 12.7 (9.0–17.6) | 21 | 53 | 14.0 (9.9–19.4) | 5 | 28 | 16.9 (10.6–25.8) | 3 | 31 | 12.7 (8.3–19.0) |
| LGV | 11 | 43 | 2.3 (1.0–5.3) | 4 | 29 | 5.1 (2.5–10.1) | 4 | 29 | 5.4 (2.6–10.6) | 1 | 18 | 5.1 (3.0–8.7) | 1 | 25 | 2.2 (1.3–3.6) |
| HSV-1 | 14 | 31 | 2.0 (0.8–4.7) | 6 | 17 | 4.1 (1.7–9.2) | 5 | 16 | 3.5 (1.4–8.3) | 1 | 7 | 12.0 (1.6–53.9) | 2 | 13 | 4.9 (0.7–27.0) |
| HD | 56 | 107 | 2.0 (1.2–3.3) | 23 | 56 | 3.5 (2.3–5.3) | 23 | 54 | 4.1 (2.7–6.3) | 3 | 24 | 3.1 (1.2–7.4) | 3 | 31 | 1.4 (0.6–3.2) |
| None | 38 | 83 | 34.7 (25.5–45.1) | 15 | 48 | 29.1 (22.0–37.4) | 14 | 44 | 38.3 (29.8–47.5) | 2 | 22 | 43.1 (33.3–53.5) | 3 | 31 | 40.3 (31.8–49.5) |

Diagnosed proportion per pathogen among populations symptomatic with (A) vaginal discharge, (B) urethral discharge, and (C) genital ulcer. Proportions are sex-matched population-weighted means with 95% CIs and were estimated using generalised linear mixed-effects model for each symptom in 3 sub-analyses: overall, stratified by HIV status, and stratified by age group.

$N_s$: number of unique studies, $N_o$: number of observations. Multiple study observations possible when outcomes were stratified by population type, country, year or time-period, sex, HIV status, or age (as available).

BV: Bacterial vaginosis, CA: *Candida albicans*, CS: *Candida* species (any), CT: *Chlamydia trachomatis*, HD: *Haemophilus ducreyi*, HSV: Herpes simplex virus (unspecified), HSV-1: Herpes simplex virus type 1, HSV-2: Herpes simplex virus type 2, LGV: lymphogranuloma venereum, MG: *Mycoplasma genitalium*, NG: *Neisseria gonorrhoeae*, TP: *Treponema pallidum*, TV: *Trichomonas vaginalis*, None: unknown aetiology.

68) of urethral discharge cases. CT was the predominant (21.9% [95% CI: 15.4% to 30.3%], $n_{UD}$ = 48) non-gonococcal cause of urethral discharge, followed by MG (5.5% [95% CI: 3.7% to 8.3%], $n_{UD}$ = 29) and TV (2.8% [95% CI: 1.8% to 4.2%], $n_{UD}$ = 46). No aetiology was detected in 10.0% (95% CI: 6.8% to 14.5%, $n_{UD}$ = 26) of cases. The odds of MG (aOR: 0.95 [95% CI: 0.91 to 1.00]) and TV (aOR: 0.93 [95% CI: 0.90 to 0.96]) decreased over time, with no significant change for NG (aOR: 1.01 [95% CI: 0.99 to 1.03]) or CT (aOR: 1.02 [95% CI: 0.99 to 1.06]; Table L in S1 Appendix). The proportion per year with unknown aetiology also decreased (aOR: 0.95 [95% CI: 0.91 to 0.99]).

In 2015, genital ulcer was predominantly caused by HSV-2 (Fig 2C and Table 2C). The diagnosed proportion of cases for HSV was 57.1% (95% CI: 40.9% to 71.9%, $n_{GU}$ = 55), HSV-2 was 48.3% (95% CI: 32.9% to 64.1%, $n_{GU}$ = 47), and HSV-1 was 2.0% (95% CI: 0.8% to 4.7%, $n_{GU}$ = 31). TP was the second most prevalent aetiology (9.3% [95% CI: 6.4% to 13.4%], $n_{GU}$ =

117), followed by LGV (2.3% [95% CI: 1.0% to 5.3%], $n_{GU}$ = 43) and HD (2.0% [95% CI: 1.2% to 3.3%], $n_{GU}$ = 107). No aetiology was identified in 34.7% (95% CI: 25.5% to 45.1%, $n_{GU}$ = 83) of cases. The distribution of genital ulcer pathogens changed substantially over time. Since most observations for HSV-1 (97%) and HSV-2 (85%) occurred after the year 2000, trends are more reliably assessed for unspecified HSV. In 1980, estimates were highest for HD (86.9% [95% CI: 79.7% to 91.8%]) and lowest for HSV (7.8% [95% CI: 4.3% to 13.7%]; Fig 2C, Table Q in S1 Appendix). The odds of HSV increased per year (aOR: 1.08 [95% CI: 1.06 to 1.11]) and decreased for HD (aOR: 0.85 [95% CI: 0.83 to 0.86]), LGV (aOR: 0.93 [95% CI: 0.89 to 0.96]), and TP (aOR: 0.98 [95% CI: 0.96 to 0.99]; Table L in S1 Appendix). The odds of having no identified aetiology (aOR: 1.02 [95% CI: 1.00 to 1.04]) increased over time. Men with genital ulcer had higher odds (aOR: 1.98 [95% CI: 1.27 to 3.08]) of HD diagnosis than women, but sex was not a significant predictor for other aetiologies.

Across all 3 symptoms, there was negligible evidence of regional variation in the estimated diagnosed proportion per pathogen over time (Fig B and Table L in S1 Appendix). Study observations were heterogeneous for all 3 symptoms, especially vaginal discharge. The percentage of total variance attributed to observation-level random effects was 43.0% for VD, 22.3% for UD, and 25.0% for GU (Table L in S1 Appendix).

## Population factors associated with the aetiology of each symptom

For each symptom, HIV status and age were significantly associated with the diagnosis of several RTIs, although the infections with the largest proportions per symptom generally remained the same for all subgroups. The main exception was for women with HIV, where vaginal discharge was predominantly caused by BV (45.5% [95% CI: 34.4% to 56.9%], $n_{VD}$ = 20), CS (31.4% [95% CI: 19.9% to 45.8%], $n_{VD}$ = 17), and TV (17.1% [95% CI: 11.7% to 24.4%], $n_{VD}$ = 22) in 2015 (Table 2A).

For vaginal discharge, the odds were higher among women with HIV than women without HIV for diagnosis with MG (aOR: 2.3 [95% CI: 1.6 to 3.4], $n_{VD}$ = 36), TV (aOR: 2.2 [95% CI: 1.6 to 2.9], $n_{VD}$ = 45), BV (aOR: 1.8 [95% CI: 1.3 to 2.5], $n_{VD}$ = 42), and NG (aOR: 1.6 [95% CI: 1.2 to 2.2], $n_{VD}$ = 41), and lower for CS (aOR: 0.5 [95% CI: 0.3 to 0.7], $n_{VD}$ = 34; Fig 3A and Table M in S1 Appendix). No HIV-stratified observations were available for CA. The odds were higher among women <25 years than ≥25 years for diagnosis with CT (aOR: 2.4 [95% CI: 1.7 to 3.3], $n_{VD}$ = 38), MG (aOR: 2.0 [95% CI: 1.3 to 3.0], $n_{VD}$ = 31), CS (aOR: 1.8 [95% CI: 1.3 to 2.7], $n_{VD}$ = 37), and NG (aOR: 1.7 [95% CI: 1.2 to 2.3], $n_{VD}$ = 43), and lower for those without an identified aetiology (aOR: 0.7 [95% CI: 0.5 to 0.9], $n_{VD}$ = 36; Fig 4A and Table N in S1 Appendix).

For urethral discharge, the odds of TV diagnosis were higher (aOR: 1.9 [95% CI: 1.1 to 3.3], $n_{UD}$ = 35) among men with HIV than men without HIV and lower for CT (aOR: 0.5 [95% CI: 0.4 to 0.8], $n_{UD}$ = 38; Fig 3B and Table M in S1 Appendix). CT was the only RTI associated with age for urethral discharge, with higher odds among men <25 years (aOR: 1.6 [95% CI: 1.2 to 2.3], $n_{UD}$ = 32; Fig 4B and Table N in S1 Appendix).

For genital ulcer, the odds of unspecified HSV (aOR: 1.8 [95% CI: 1.1 to 2.8], $n_{GU}$ = 53) and HSV-2 (aOR: 1.6 [95% CI: 1.1 to 2.5], $n_{GU}$ = 57) were higher among participants with HIV than those without HIV, but lower for those with unidentified aetiology (aOR: 0.7 [95% CI: 0.5 to 0.9], $n_{GU}$ = 92; Fig 3C and Table M in S1 Appendix). LGV (aOR: 2.4 [95% CI: 1.3 to 4.5], $n_{GU}$ = 43) and HD (aOR: 2.3 [95% CI: 1.3 to 4.1], $n_{GU}$ = 55) had higher odds of diagnosis among participants <25 years (Fig 4C and Table N in S1 Appendix). In both analyses, men with genital ulcer had lower odds (aOR: 0.4 [95% CI: 0.3 to 0.7]; aOR: 0.6 [95% CI: 0.4 to 1.0]) of diagnosis with HSV-2 than women (Tables M and N in S1 Appendix).

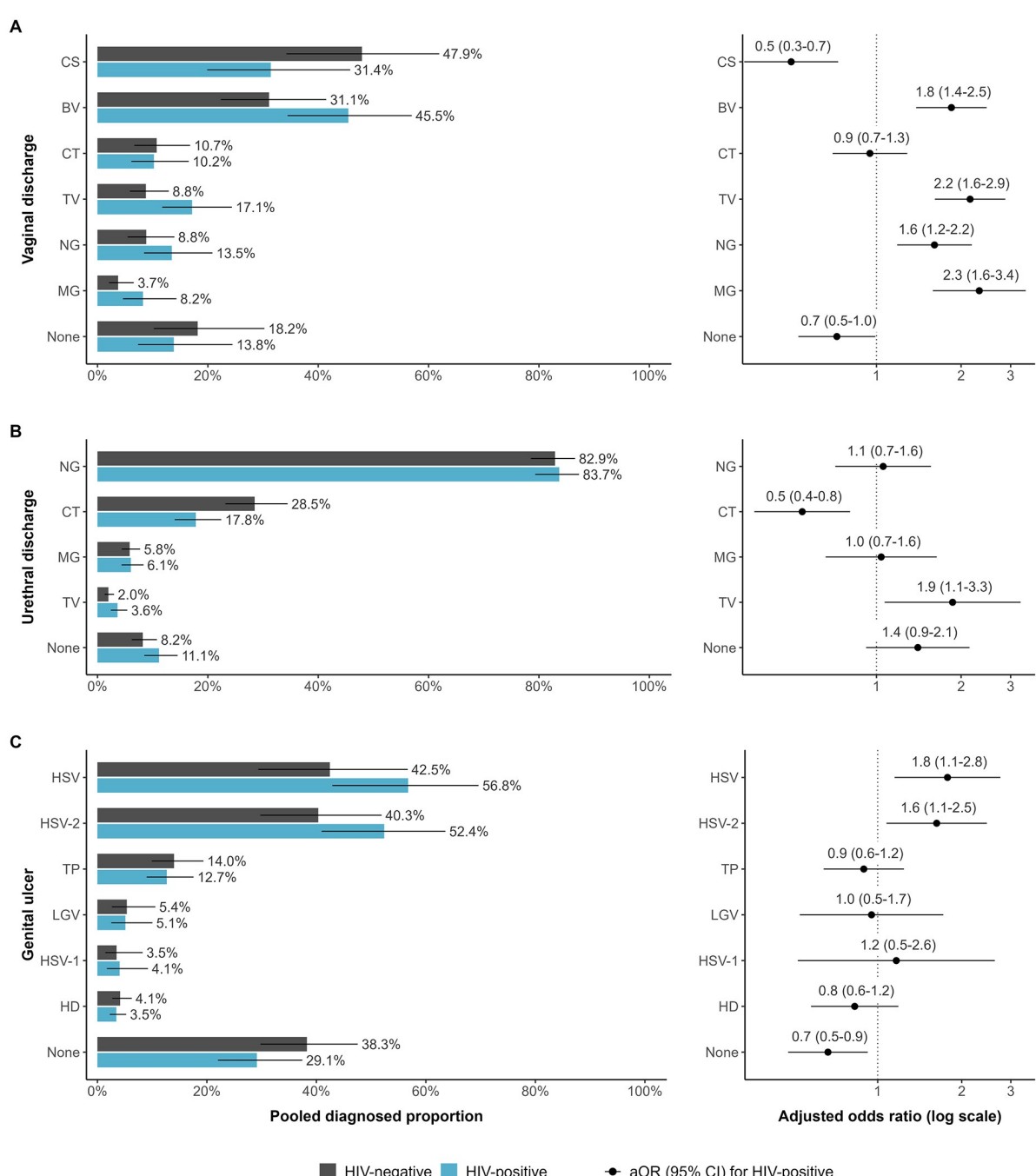

**Fig 3. Comparison of diagnosed proportion per pathogen by HIV status in sub-Saharan Africa.** Estimated diagnosed proportion per pathogen among adults with and without HIV in 2015 (left) and adjusted odds of diagnosis per pathogen among adults with HIV compared to adults without HIV (right) symptomatic with (A) vaginal discharge, (B) urethral discharge, and (C) genital ulcer. Proportions and odds estimated using generalised linear mixed-effects model for each symptom with fixed effects for RTI and the interaction of RTI with year, HIV status, and sex (genital ulcer only), random intercepts for the interaction of RTI and region, and observation-level random intercepts. Bars represent sex-matched population-weighted mean proportions in 2015. Points represent the adjusted odds of diagnosis among adults with HIV. Solid lines represent 95% CIs. BV: Bacterial vaginosis, CS: *Candida* species (any), CT: *Chlamydia trachomatis*, HD: *Haemophilus ducreyi*, HSV: Herpes simplex virus (unspecified), HSV-1: Herpes simplex virus type 1, HSV-2: Herpes simplex virus type 2, LGV: lymphogranuloma venereum, MG: *Mycoplasma genitalium*, NG: *Neisseria gonorrhoeae*, TP: *Treponema pallidum*, TV: *Trichomonas vaginalis*, None: unknown aetiology.

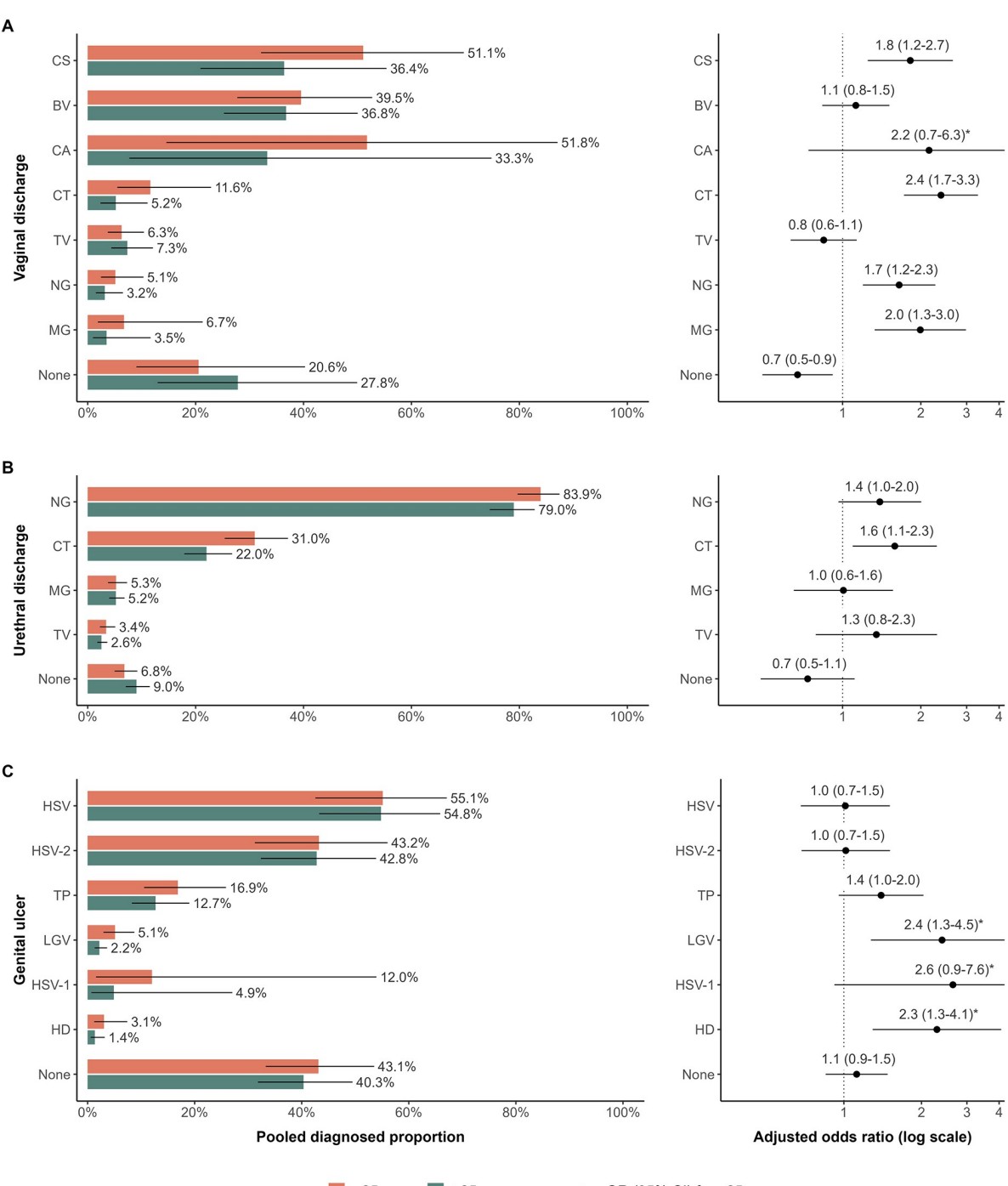

**Fig 4. Comparison of diagnosed proportion per pathogen by age group in sub-Saharan Africa.** Estimated diagnosed proportion per pathogen among youth <25 years and adults ≥25 years in 2015 (left) and adjusted odds of diagnosis per pathogen among youth compared to adults (right) symptomatic with (A) vaginal discharge, (B) urethral discharge, and (C) genital ulcer. Proportions and odds estimated using generalised linear mixed-effects model for each symptom with fixed effects for RTI and the interaction of RTI with year, age group, and sex (genital ulcer only), random intercepts for the interaction of RTI and region, and observation-level random intercepts. Bars represent sex-matched population-weighted mean proportions in 2015. Points represent the adjusted odds of diagnosis among youth. Solid lines represent 95% CIs. *Confidence intervals truncated by x-axis. BV: Bacterial vaginosis, CA: *Candida albicans*, CS: *Candida* species (any), CT: *Chlamydia trachomatis*, HD: *Haemophilus ducreyi*, HSV: Herpes simplex virus (unspecified), HSV-1: Herpes simplex virus type 1, HSV-2: Herpes simplex virus type 2, LGV: lymphogranuloma venereum, MG: *Mycoplasma genitalium*, NG: *Neisseria gonorrhoeae*, TP: *Treponema pallidum*, TV: *Trichomonas vaginalis*, None: unknown aetiology.

## Critical appraisal and sensitivity analyses

Fewer than half of studies aimed to assess the aetiology of genital symptoms ($N_{VD}$ = 37/87, $N_{UD}$ = 32/55, $N_{GU}$ = 40/80, Tables O and P in S1 Appendix). Most studies were at risk of selection bias; participant inclusion criteria were often clearly defined ($N_{VD}$ = 82/87, $N_{UD}$ = 50/55, $N_{GU}$ = 73/80), but few studies recruited participants using consecutive or random sampling methods ($N_{VD}$ = 32/87, $N_{UD}$ = 27/55, $N_{GU}$ = 46/80), had adequate participation rates ($N_{VD}$ = 34/87, $N_{UD}$ = 21/55, $N_{GU}$ = 40/80), or provided sufficient detail on participant characteristics to determine their representativeness ($N_{VD}$ = 36/87, $N_{UD}$ = 21/55, $N_{GU}$ = 56/80). Most studies were not susceptible to measurement bias; the majority defined participant symptoms objectively ($N_{VD}$ = 48/87, $N_{UD}$ = 36/55, $N_{GU}$ = 75/80), employed consistent diagnostic methodologies ($N_{VD}$ = 81/87, $N_{UD}$ = 47/55, $N_{GU}$ = 75/80), and avoided misclassifying infections by testing for multiple pathogens ($N_{VD}$ = 43/87, $N_{UD}$ = 22/55, $N_{GU}$ = 48/80). Studies generally had sample sizes of at least 100 participants ($N_{VD}$ = 61/87, $N_{UD}$ = 39/55, $N_{GU}$ = 46/80) and reported results unambiguously ($N_{VD}$ = 51/87, $N_{UD}$ = 37/55, $N_{GU}$ = 52/80). Several of these appraisal criteria were associated with the odds of RTI diagnosis, particularly for vaginal discharge, but the hierarchy of aetiologies per symptom remained the same (Fig C in S1 Appendix).

In sensitivity analyses, estimates derived for RTIs with large diagnostic test performance adjustments (CS, BV, TV, and NG) were higher than those based on reported observations, although trends were relatively similar (Fig D and Table Q in S1 Appendix). Estimates derived using NAAT only were generally consistent with trend estimates based on all test types, irrespective of diagnostic test performance adjustment (Fig E in S1 Appendix). Notable exceptions when restricted to NAAT-based estimates were for vaginal discharge, where TV was stable and NG increased over time, and for genital ulcer, where TP increased over time. However, NAAT-based estimates were interpolated from shorter time spans than estimates derived using all diagnostic tests.

## Discussion

This systematic review and meta-regression estimated the distribution, trends, and determinants of aetiologies for STI-related symptoms in sub-Saharan Africa from 1970 to 2022. The primary aetiologies identified for vaginal discharge were candidiasis, bacterial vaginosis, and trichomoniasis, for urethral discharge were gonorrhoea and chlamydia, and for genital ulcer were herpes and syphilis. Additionally, over 5% of vaginal discharge and urethral discharge cases were attributed to *M. genitalium*, and over a quarter of vaginal discharge and genital ulcer cases lacked an identified cause in 2015. Symptom aetiologies changed over time, particularly for genital ulcer where the leading cause shifted from chancroid in 1990 to HSV-2 by 2010. We did not find evidence of systematic regional variation in the aetiologic distributions. Although HIV status and age group were significantly associated with the diagnosis of several RTIs, the overall hierarchy of aetiologies for each symptom largely remained unchanged.

The estimated distribution of aetiologies for each symptom was consistent with infections prioritised by WHO syndromic management guidelines [3]. Our analysis adds to previous reviews [31–35] that informed WHO syndromic algorithms by incorporating a wider range of studies reporting underlying syndrome aetiologies, specifically in SSA. We, however, did not find evidence for needing to adapt the algorithms for SSA settings or populations. Guidelines currently recommend assessing for *M. genitalium* only in instances of recurrent or persistent discharge [3], which would potentially leave this infection untreated. Additional attention to MG is therefore warranted, particularly amid ongoing debates on aetiologic testing in higher-

income settings [36]. The high proportion of cases without an identifiable cause poses a persistent unaddressed treatment challenge, even with expanded access to diagnostic testing.

Changes over time in genital ulcer aetiology are consistent with existing evidence indicating the near eradication of chancroid [37]. The reduction in chancroid cases is largely attributed to the introduction of antibiotics for genital ulcer management. The prevalence of HSV-2 has also declined, only much more slowly [38]. Furthermore, the HIV epidemic has also shaped the epidemiology of ulcerative infections over time, with documented synergy between HSV-2 and HIV [39]. We identified that 2% and 48% of genital ulcer cases were attributed to HSV-1 and HSV-2 in 2015, respectively, consistent with other systematic review findings of 1.2% and 51% in SSA during 1990 and 2015 [38,40].

These temporal changes underscore the need for regular aetiologic assessment of syndromes. However, among 32 countries with data in our review, the publication rate approximated 1 study every 10 years (median of 5 studies per country during 1969 and 2022), falling short of the WHO's recommended assessment frequency of 2 years [6]. In the absence of local studies, our findings suggest data from neighbouring regions or countries can be used to inform syndromic management protocols, although national-level monitoring of antimicrobial resistance for key pathogens may still be necessary [41].

Aetiologic proportions among symptomatic populations are a component of characterising RTI burden, but these metrics are not equivalent. Diagnosed proportions reflect the prevalence of all possible aetiologies for a particular symptom, conditional on individuals being symptomatic and seeking care. For example, while HSV-2 prevalence has decreased in the region, the proportion of genital ulcer cases attributed to HSV-2 increased over time because the prevalence of chancroid and syphilis declined more rapidly [38,42–44]. Decreasing aetiologic proportions for syphilis and chancroid were consistent with general population prevalence trends in SSA [16,42–45]. Diagnosed proportions also do not reflect the relative RTI burden among different population groups. Although aetiologic proportions were distributed similarly by HIV status, age group, and sex (genital ulcer) for each symptom, the prevalence of RTIs in SSA is higher among women and people living with HIV, and varies by age [46,47]. Using our results to estimate overall STI burdens requires additional data on the prevalence of STI syndromes, care seeking, and aetiologies, collected from representative samples of the general population.

Our analysis had limitations, mainly related to features of the studies and data. We extensively searched grey literature to identify all available data, particularly surveillance reports that might not appear in peer-reviewed academic literature. The grey literature search used similar criteria to our database search but was not strictly systematic. We did not find any studies in 16 countries, particularly in central Africa (5/9 countries had no studies). However, the consistency of results across countries with data provides some confidence that key syndromic management recommendations are generalisable to countries that did not have any or recent studies. Most studies were not nationally representative and were conducted among a convenience sample of symptomatic individuals seeking healthcare at specific facilities. Our analysis did not account for factors influencing treatment access, such as urban or rural location [48–50], or symptom severity, recurrence, or treatment history, which could have influenced the aetiologic distributions among study participants but for which data were not consistently available. Additionally, as most studies did not report HIV prevalence or age among symptomatic participants, we were unable to adjust for these factors in the overall estimates. These factors may have contributed to heterogeneity in the study observations, which was particularly large for vaginal discharge. Our definition of vaginal or urethral discharge based on abnormal symptoms may have captured a subset of those meeting criteria for diagnosis with vaginal discharge syndrome (abnormal discharge, vulval irritation, and/or itching) or

urethral discharge syndrome (abnormal discharge, dysuria, and/or itching). Alternatively, our symptom-based definition may have included more individuals with non-infectious aetiologies than a clinical syndromic diagnosis. The symptom-based definition was chosen to accommodate variations in reporting across studies and over time but may have influenced estimated aetiologic distributions. Several studies did not differentiate between HSV type or Candida species; we considered these data representative of HSV-2 and CA trends, which may have overestimated their contribution. Our estimated diagnosed proportion for HSV-2 was 85% that of unspecified HSV, which was consistent with previous studies [38]. In contrast, the diagnosed proportion for CA was 45% relative to CS, which was below the expected range of 70% to 90% [51,52]. Adjustments to the performance of gram stain and/or wet mount may have overestimated CS proportions, while CA may have been underestimated due to the limited number of observations. Other RTI proportions may also have been over- or underestimated due to assigned diagnostic test sensitivity and specificity values, despite our efforts to ensure their accuracy. Our modelling approach and estimates did not account for relationships between similar outcomes, such as the contribution of HSV-1 and HSV-2 to unspecified HSV; coinfection rates between RTIs; or the negative correlation between proportions for the same symptom, which depend on coinfection rates. Finally, due to different numbers of pathogens examined across studies, we were required to subjectively define an unknown aetiology among study participants tested for 3 or more RTIs, irrespective of the pathogens examined or the number of potential pathogens present. Despite these limitations, to the best of our knowledge, our study provides the most comprehensive analysis to date on the aetiology of STI-related symptoms in SSA.

In conclusion, the aetiology of 3 common STI-related symptoms were remarkably similar across regions in sub-Saharan Africa but have evolved over time, underscoring a changing STI transmission landscape and the need for regular re-assessment to inform syndromic management protocols. The observed aetiologic distributions in SSA were largely consistent with WHO recommended syndromic management algorithms without strong evidence of variation by country, context, or population strata, strengthening the generalisability of our findings to settings lacking data in SSA.

## Supporting information

**S1 Appendix. Supporting information.**
(PDF)

**S2 Appendix. Data used in the analysis.**
(XLSX)

**S1 Checklist. Preferred reporting items for systematic reviews and meta-analyses (PRISMA) checklist.**
(PDF)

## Author Contributions

**Conceptualization:** Julia Michalow, Anne Cori, Marie-Claude Boily, Jeffrey W. Imai-Eaton.

**Data curation:** Julia Michalow, Magdalene K. Walters, Olanrewaju Edun, Max Wybrant, Bethan Davies, Tendesayi Kufa, Thabitha Mathega, Sungai T. Chabata, Frances M. Cowan.

**Formal analysis:** Julia Michalow.

**Supervision:** Jeffrey W. Imai-Eaton.

**Visualization:** Julia Michalow.

**Writing – original draft:** Julia Michalow.

**Writing – review & editing:** Julia Michalow, Magdalene K. Walters, Olanrewaju Edun, Max Wybrant, Bethan Davies, Tendesayi Kufa, Thabitha Mathega, Sungai T. Chabata, Frances M. Cowan, Anne Cori, Marie-Claude Boily, Jeffrey W. Imai-Eaton.

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
