## [Editor Report · Decision Letter 0]

6 Nov 2023

Dear Dr Michalow, 

Thank you for submitting your manuscript entitled "Aetiology of vaginal discharge, urethral discharge, and genital ulcer in sub-Saharan Africa: systematic review and meta-regression" for consideration by PLOS Medicine.

Your manuscript has now been evaluated by the PLOS Medicine editorial staff as well as by an academic editor with relevant expertise and I am writing to let you know that we would like to send your submission out for external peer review.

Please re-submit your manuscript within two working days, i.e. by Nov 08 2023 11:59PM.

Feel free to email me at lgaynor@plos.org if you have any queries relating to your submission.

Kind regards,

Louise Gaynor-Brook, MBBS PhD

---

## [Decision Letter · Decision Letter 1]

19 Dec 2023

Dear Dr. Michalow,

Many thanks for submitting your manuscript "Aetiology of vaginal discharge, urethral discharge, and genital ulcer in sub-Saharan Africa: systematic review and meta-regression" (PMEDICINE-D-23-03176R1) for consideration at PLOS Medicine. 

Your paper has been evaluated and discussed by our editorial team along with an Academic editor with relevant expertise. The paper generated a substantial amount of interest among reviewers, and we have obtained comments from five subject reviewers and a statistical reviewer. The reviews are appended at the bottom of this email and any accompanying reviewer attachments can be seen via this link: [LINK]

As you will see, the reviewers were overall very supportive of the study and felt it was well done, but they had a number of specific questions and concerns. In light of the reviews, I’m pleased to invite you to submit a revised manuscript that addresses the reviewers' and editors' comments. Of course, at this stage we cannot offer any guarantees regarding publication, and we plan to seek re-review by one or more of the reviewers. 

Our usual deadline for resubmission is 3 weeks; however, in view of the impending holidays and our editorial requests, we are happy to allow you more time than this. However, it would be useful if you could email me (hvanepps@plos.org) or my colleague Louise (the handling editor; lgaynor@plos.org) to give us a rough idea of when you think it will be feasible to resubmit. 

We look forward to receiving your revised manuscript. 

Kind Regards,

Heather

Heather Van Epps, PhD

Executive Editor

[on behalf of]

Louise Gaynor-Brook, MBBS PhD

PLOS Medicine

plosmedicine.org

1. Per PLOS Medicine policy, we ask that you update your literature to within 6 months of the present time (minimally 6 months before submission);

2. At this stage, we ask that you include a short, non-technical Author Summary of your research to make findings accessible to a wide audience that includes both scientists and non-scientists. The Author Summary should immediately follow the Abstract in your revised manuscript. This text is subject to editorial change and should be distinct from the scientific abstract. Ideally each sub-heading should contain 2-3 single sentence, concise bullet points containing the most salient points from your study. In the final bullet point of ‘What Do These Findings Mean?’ Please include the main limitations of the study in non-technical language. Please see our author guidelines for more information: https://journals.plos.org/plosmedicine/s/revising-your-manuscript#loc-author-summary.

3. When you submit your revised paper, please upload any figures associated with the paper as individual TIF or EPS files with 300dpi resolution at resubmission; please read our figure guidelines for more information on our requirements: http://journals.plos.org/plosmedicine/s/figures. While revising your submission, please also upload your figure files to the PACE digital diagnostic tool, https://pacev2.apexcovantage.com/. PACE helps ensure that figures meet PLOS requirements. To use PACE, you must first register as a user. Then, login and navigate to the UPLOAD tab, where you will find detailed instructions on how to use the tool. If you encounter any issues or have any questions when using PACE, please email us at PLOSMedicine@plos.org.

4. When you submit your revision, please re-upload the PRISMA checklist using line numbers (rather than section alone) to indicate the relevant position of information in the paper;

5. We ask every co-author listed on the manuscript to fill in a contributing author statement, making sure to declare all competing interests. If any of the co-authors have not filled in the statement, we will remind them to do so when the paper is revised. If all statements are not completed in a timely fashion this could hold up the re-review process. If new competing interests are declared later in the revision process, this may also hold up the submission. Should there be a problem getting one of your co-authors to fill in a statement we will be in contact. Please ensure that you do not add or remove authors without first discussing this and gaining agreement from the handling editor. You can see our competing interests policy here: http://journals.plos.org/plosmedicine/s/competing-interests.

6. Minor: Please place references inside punctuation (Vancouver style);

Academic Editor comments:

1. There is a lack of attention to heterogeneity between studies: the very informative Figure 1 indicates that there is substantial heterogeneity, but this needs to be quantified. What was the between-study variance (tau squared)? How much of the variance was explained by the variables included in the meta-regression? Which variable explained the most?

2. The assessment of risk of bias is not appropriate. The JBI instrument does not assess the risk of bias; rather, it is a more general measure that mixes risk of bias criteria with reporting and general study quality. Several criteria used have nothing to do with the risk of bias (for example, 1, 7, 10). Further, summary scores should be avoided: they involve inappropriate equal weighting of the different criteria. See also Dekkers et al. COSMOS-E statement. PLoS Med 2019. My advice would be to analyse the criteria related to bias (3, 4, 5, 8) separately in the meta-regression. It would also be interesting to compare the convenience sampling studies with those using consecutive or random sampling.

3. I felt the authors tend to overinterpret their findings. For example, the data do not support the recommendation that a survey should only be done every 5 years. Rather, the authors should stress that studies are urgently needed in many countries without data. Also, the limitations of the syndromic approach should be discussed. An interesting question in this context relates to the proportion of patients who will not receive adequate treatment based on their results.

Comments from the reviewers:

Reviewer 1:

Thank you for the opportunity to review this manuscript. This is a valuable and timely study. It is delightful to see this study conducted, and so rigorously. The study is thorough, well-conducted, and conforms to the best standards in conducting systematic reviews. The article is well-written, and the presentation is lucid. The results are of global interest and inform the current intense discussion about the relevance of the syndromic approach for the management of STIs versus the etiological approach. The results directly inform regional (as well as global) guidelines.

In the spirit of enhancing the impact and value of this study, I suggested few minor revisions.

1. The definition of the proportion of cases that did not have an identified etiology was unclear to me. I assume the proportion depends on what is being tested in each study, but what is being tested can differ between studies. This makes this proportion not well-defined. Can you please clarify and discuss this point?

2. There is a challenge in applying adjustments for the diagnostic test performance. The reason is that the existing reported adjustments tend to be not representative, introducing errors (sometimes even non-real negative values) when applied. It is great that the authors presented results with and without these adjustments, and both overall agreed. I think it might still be useful to discuss limitations of these adjustments in the limitations section.

3. The results for HSV-1 are interesting on their own but are not sufficiently discussed. The results also align with another relatively recent study (Harfouche M, Chemaitelly H, Abu-Raddad LJ. Herpes simplex virus type 1 epidemiology in Africa: Systematic review, meta-analyses, and meta-regressions. J Infect 2019; 79(4): 289-99.). I suggest some discussion here given the increasing relevance of this infection as an STI (though not strictly in Africa).

4. Although such studies are rare, why did you include studies with participants as young as 10 years? It seems to me that a more appropriate age threshold is 15 years. You may want to justify your choice.

5. You may want to indicate in the limitations that the search for grey literature was not strictly systematic.

Reviewer 2: 

This article represents a truly massive effort to use the existing literature to explore the cause of vaginal and penile discharge and genital ulcers in SSA from 1969 to 2022. The authors recognize a variety of limitations in this effort. The most important problem with the article lies in failure to discuss the reasons for some of the striking changes offered in the figures: huge increase in NGC, virtual disappearance of HD, reduction in TV and many, many more observations. For example, did untreated HIV set the stage for spread of HD, that has abated with ART? The purpose of the Discussion is to INTERPRET results, not reiterate them.

Comments:

1. The authors state: "STI surveillance using syndrome-based assessments is noncomprehensive and requires studies among symptomatic and asymptomatic populations." Did they not find articles that offered a view of unrecognized, untreated asymptomatic infection? For example guidelines call for routine screening of people living with HIV for STIs?

2. In the Methods the authors say "we searched from inception to 25 July 2022. I think by inception they mean 1969?

3. The authors discuss bias " Most studies (NVD=47, NUD=30, NGU=33) had moderate risk of bias (Table 1, Table S17). Studies with higher risk of bias (NVD=19, NUD=11, NGU=12) were predominantly those with alternate study objectives, insufficient description of study participants and/or settings, only one pathogen assessed, and ambiguous reporting of outcomes. Estimates for the proportion diagnosed per pathogen over time were generally consistent when alternatively including studies of any risk level, or only studies with lower and/or moderate risk of bias (Figure S3)." . I am not sure they are describing bias (in the epidemiology sense) as much as limitations of the articles available and the veracity of sampling and tests employed? The idea of what they are trying to do could be stated more clearly.

4. The conclusion of the article does not fit: The authors state "STI surveillance requires prevalence studies among both symptomatic and asymptomatic populations, particularly due to high rates of asymptomatic infection". But this article is NOT about asymptomatic infections. That is an entirely different topic. The authors need to think of what the reader might take from this work? They implore more frequent surveillance but I am not sure the data support this idea without further explanation of the changes observed and more consideration of the frequency if this effort?

5. Most important, the authors do not offer a clear opinion of syndromic management compared to diagnostic results. This descriptive effort surely is designed to better direct syndromic management which remains the mainstay of STD care. Is that the intention of the authors? On the other hand (for example) focus on treatment of GC, found so commonly in discharge, would leave the far less common MG untreated. It seems unwise to put forth this effort without an opinion about this dilemma, and the potential contribution of this report?

Reviewer 3 (statistics):

Firstly, I would like to commend the authors on the substantive amount of work undertaking this systematic review. There is a large number of included studies, and I can appreciate the workload behind such a task. Just to be clear, I will only be considering the methods and statistical analyses undertaken. Overall, I believe there is good methodological and scientific rigour utilised throughout the review. The analysis is, for the most part, well explained and is conducted well. I have some points below for the authors consideration.

1. The authors have presented a lot of details regarding the meta-regression modelling. However, there appears to be a lack of information regarding the initial pairwise meta-analysis which led to this. While there are very helpful and detailed figures (e.g. Figure 1, although I think this is misnamed as the PRISMA flow diagram is also named Figure 1) show the aOR by vaginal discharge, urethral discharge, and genital ulcer they do show that, in some cases, there may be high heterogeneity. For example, figure 1, vaginal discharge shows heterogeneity between the groups. There appears to be a lack of quantification of this heterogeneity (i.e. tau: the standard deviation between the studies). Please could the authors report this information. Additionally, it would be beneficial to observe how meta-regression addressed such heterogeneity and which factors were significantly associated with the observed heterogeneity. 

2. The risk of bias assessment includes an overall assessment, which is not recommended. The authors should consider removing this information. Currently, the risk of bias assessment gives equal weighting to each item and some of the items are to do with the study quality of reporting and not risk of bias per se. Therefore, the inclusion of the overall risk of bias score in an analysis would be misleading, and it would be more pertinent to group studies based on certain risk of bias questions as categorical outcome (yes, no, unclear). For example, questions 3, 4, 5, and 8. Additionally, the sampling utilised in the studies should also be considered (i.e. compare convenience sampling studies with those using consecutive or random sampling methods). 

These re-analyses may change the interpretation of the results and this should be considered carefully. Currently, the results are interpreted positively and maybe too positively. 

3. PRISMA: Number of reports included was 198 but when adding the subgroups there are 227, please check and make clear. 

4. Text S2: In the supplementary text the authors state that in the case of multiple sources being available with wide variation, a mean sensitivity and specificity value was calculated. I believe taking the mean would be inadequate, as the variation most likely occurs due to sample size differences and other confounding factors (e.g. high vs low risk population). It would therefore be beneficial for these values to be created using a weighted mean based on the sample size.

5. The tables need to make clear that they are referring the number of studies and not number of participants. 

6. There is a lack of information regarding how the adjusted odds ratio were calculated and what factors were used. Please elaborate. A similar supplementary text as to the other analyses would be beneficial. 

Reviewer 4:

This manuscript by Michalow et al. describes a systematic review and meta-regression to characterise aetiologies for vaginal discharge, urethral discharge, and genital ulcer in sub-Saharan Africa. International and national guidelines for STIs across Africa are often informed by sporadic studies and surveillance activities, making amalgamation of this information very helpful. The manuscript itself is clear an

---

## [Decision Letter · Decision Letter 2]

13 Mar 2024

Dear Dr. Michalow,

Thank you very much for re-submitting your manuscript "Aetiology of vaginal discharge, urethral discharge, and genital ulcer in sub-Saharan Africa: systematic review and meta-regression" (PMEDICINE-D-23-03176R2) for review by PLOS Medicine.

I have discussed the paper with my colleagues and the academic editor and it was also seen again by three reviewers. I am pleased to say that provided the remaining editorial and production issues are dealt with we are planning to accept the paper for publication in the journal.

[LINK]

We expect to receive your revised manuscript within 1 week. Please email me (lgaynor@plos.org) if you have any questions or concerns.

We look forward to receiving the revised manuscript by Mar 20 2024 11:59PM.   

Sincerely,

Louise Gaynor-Brook, MBBS PhD

plosmedicine.org

lgaynor@plos.org

Requests from Editors:

Thank you for your patience with a longer assessment process than we anticipated, and apologies for the delay in providing you with an editorial decision. The list below appears rather lengthy, but some of these points are more minor points which should not require a substantial amount of time to attend to. 

General comments:

Throughout the paper, please adapt reference call-outs to the following style: "... every year [1,2]." (noting the absence of spaces within the square brackets).

Please use person-first language throughout your manuscript e.g. ‘men with HIV’ rather than “HIV-positive men”, including in figure legends

When referring to age, please revise to e.g. “The odds were higher among women <25 years than ≥25 years for diagnosis…” 

To help us extend the reach of your research, please provide any Twitter handle(s) that would be appropriate to tag, including your own, your coauthors’, your institution, funder, or lab.

Title: Please revise your title to “...Saharan Africa: A systematic review and meta-regression”

Abstract:

Please ensure that your abstract is reported according to PRISMA for abstracts, following the PLOS Medicine abstract structure (Background, Methods and Findings, Conclusions): http://www.plosmedicine.org/article/info:doi/10.1371/journal.pmed.1001419

Please provide further detail on the dates of search (from inception), types of study designs included, eligibility criteria, and synthesis/appraisal methods. 

Please define CI at first use.

Please add “95% CI” to each of the square brackets for clarity.

Please define HSV at first use.

In the last sentence of the Abstract Methods and Findings section, please describe 2-3 of the main limitations of the study's methodology.

Please begin your Abstract Conclusions with "In this study, we observed ..." or similar, to summarize the main findings from your study, without overstating your conclusions. Please emphasize what is new and address the implications of your study, being careful to avoid assertions of primacy. 

Please define WHO at first use.

Author Summary:

Thank you for providing an Author Summary. 

Lines 48 & 68 - please revise ‘Syndromic case management’ for clarity for a broad readership.

Line 69 - please define STI at first use.

In the final bullet point of ‘What Do These Findings Mean?’, please describe the main limitations of the study in non-technical language.

Methods:

Please ensure to identify any changes in the analysis (including those made in response to peer review comments) in the Methods section of the paper, with rationale. 

Please add the following statement, or similar, to the Methods: "This study is reported as per the Preferred Reporting Items for Systematic Reviews and Meta-Analyses (PRISMA) guideline (S1 Checklist)." Two copies of the PRISMA checklist have been provided - please only include the version currently included as Table S9. This should be a standalone file relabelled “S1 checklist”

Results: 

Line 258 onwards: Please add “95% CI” to each of the brackets for clarity.

Line 266 - please define aOR at first use.

Line 306 - please revise to “The odds were higher among women <25 years than ≥25 years for diagnosis…” 

Discussion:

Line 349 - Should this be “1970 to 2023”?

Please present and organize the Discussion as follows: a short, clear summary of the article's findings; what the study adds to existing research and where and why the results may differ from previous research; strengths and limitations of the study; implications and next steps for research, clinical practice, and/or public policy; one-paragraph conclusion.

Line 368 - please revise to ‘...no evidence for…’

Line 441 - please temper assertions of primacy by adding ‘to the best of our knowledge’ or similar 

Please remove the information on competing interests, funding and data sharing from the

end of the main text. This information will appear in the article metadata, via entries in the submission form.

Figures:

Fig 3, 4 - Please indicate which factors are adjusted for in the figure legend

Please define abbreviations used in the figure legend of each figure, including in the supplementary information.

Please consider avoiding the use of red and green in order to make your figure more accessible to those with colour blindness.

Tables:

Please define abbreviations used in the table legend of each table, including in the supplementary information.

Supplementary tables - Where aOR are presented, please indicate which factors are adjusted for in the respective table legend

When a p value is given, please specify the statistical test used to determine it. When reporting p values please report as p<0.001 and where higher as the exact p value e.g. p=0.002.

Please provide the unadjusted comparisons as well as the adjusted comparisons.

References:

Please ensure that journal name abbreviations match those found in the National Center for Biotechnology Information (NCBI) databases (http://www.ncbi.nlm.nih.gov/nlmcatalog/journals), and are appropriately formatted and capitalised. Please also see https://journals.plos.org/plosmedicine/s/submission-guidelines#loc-references for further details on reference formatting. 

Comments from Reviewers:

Reviewer #1: Thank you for considering and addressing my comments. 

Reviewer #3: I appreciate the hard work that has gone into updating this review. The authors have been very accommodating to the reviewers and made substantial changes that have improved the manuscript. 

Reviewer #5: Most comments I had made have been answered appropriately. Two minute crrections/additions:

I would simply nuance the statement made by the authors re use of Darkfield microscopy for TP. I could not find the statement of high sensitivity of nearly 100% in the WHO STI Laboratory manual 2023 -- at best the authors of the chapter on Syphilis stated that DF should be reserved to specialist laboratories and experienced readers. only there can sensitivity be so high. So add - ....in experienced microscopists somewhere.

They somehow did not state what they considered syphilis positive when no pathogen were recovered but syphilis serology results were available -- did they go with the interpretation made by the original authors, or did they define themselves active syphilis, and if so how? (this should be mentioned in the paper)

The authors have replaced antibody test for CT by ELISA. I still assume they talk about antibody or antigen detection system, eg direct immunofluorescent assay (DFA) or Ag detection test, again as described in the relevant chapter of the WHO Lab manual 2023 -- surely no one is making a diagnosis of CT by serological ELISA

[LINK]

---

## [Editor Report · Decision Letter 3]

21 Mar 2024

Dear Dr. Michalow,

Thank you very much for re-submitting your manuscript "Aetiology of vaginal discharge, urethral discharge, and genital ulcer in sub-Saharan Africa: A systematic review and meta-regression" (PMEDICINE-D-23-03176R3) for review by PLOS Medicine.

I am pleased to say that provided the remaining editorial and production issues are dealt with, we are planning to accept the paper for publication in the journal. The remaining editorial requests that need to be addressed are listed at the end of this email. Please take these into account before resubmitting your manuscript:

[LINK]

In revising the manuscript for further consideration here, please ensure you address the specific points made by the editors. In your rebuttal letter you should indicate your response to the reviewers' and editors' comments and the changes you have made in the manuscript. Please submit a clean version of the paper as the main article file. A version with changes marked must also be uploaded as a marked up manuscript file.

We expect to receive your revised manuscript within 1 week. Please email me (lgaynor@plos.org) if you have any questions or concerns.

We look forward to receiving the revised manuscript by Mar 28 2024 11:59PM.   

Sincerely,

Louise Gaynor-Brook, MBBS PhD

Senior Editor

PLOS Medicine

lgaynor@plos.org

plosmedicine.org

Requests from Editors:

Thank you for sharing the code that underpins the findings in your manuscript. Because Github depositions can be readily changed or deleted, please make a permanent DOI copy (e.g. in Zenodo) and provide this URL for this in your Data Availability Statement.

In your Abstract, please provide further detail on the types of study designs included (e.g. cohort studies, etc), and methods for assessing risk of bias. Please ensure that all numbers presented are identical to numbers presented in the main manuscript text.

Throughout the Results section, please indicate clearly where the full results are shown (table or figure numbers) for the results presented in the main text. Please ensure that all numbers presented in main text are identical to numbers presented in the tables.

Line 256 - Where is it shown that "studies were predominantly in South Africa"? 

Line 265: For results relating to sample size of 100 or more, NGU differs between the main text and Table 1. Please clarify.

Please incorporate Table S15 into the main paper. 

Please refer also to Table S12C in the paragraph relating to genital ulcers.

Line 301 - Where is it shown that "In 1980, estimates were highest for HD"?

Line 340 - Where is it shown that "men with genital ulcer had lower odds (aOR: 0.4 [95% CI: 0.3-0.7]; aOR: 0.6 [95% CI: 0.4-1.0]) of 341 diagnosis with HSV-2 than women"?

[LINK]

---

## [Editor Report · Decision Letter 4]

26 Mar 2024

Dear Dr Michalow, 

On behalf of my colleagues and the Academic Editor, Prof. Matthias Egger, I am pleased to inform you that we have agreed to publish your manuscript "Aetiology of vaginal discharge, urethral discharge, and genital ulcer in sub-Saharan Africa: A systematic review and meta-regression" (PMEDICINE-D-23-03176R4) in PLOS Medicine.

PRESS

Sincerely, 

Louise Gaynor-Brook, MBBS PhD 

Senior Editor 

PLOS Medicine